# Beta 2-Adrenergic Receptor in Circulating Cancer-Associated Cells Predicts for Increases in Stromal Macrophages in Circulation and Patient Survival in Metastatic Breast Cancer

**DOI:** 10.3390/ijms23137299

**Published:** 2022-06-30

**Authors:** Kirby P. Gardner, Massimo Cristofanilli, Saranya Chumsri, Rena Lapidus, Cha-Mei Tang, Ashvathi Raghavakaimal, Daniel L. Adams

**Affiliations:** 1School of Graduate Studies, Rutgers University, Piscataway, NJ 08901, USA; 2Creatv MicroTech, Inc., 9 Deer Park Dr., Monmouth Junction, NJ 08852, USA; dan@creatvmicrotech.com; 3Sandra and Edward Meyer Cancer Center, Weill Cornell Medicine, New York, NY 10065, USA; mac9795@med.cornell.edu; 4Jacoby Center for Breast Health, Mayo Clinic Cancer Center, Jacksonville, FL 32224, USA; chumsri.saranya@mayo.edu; 5Greenebaum Cancer Center, School of Medicine, University of Maryland, Baltimore, MD 21201, USA; rlapidus@som.umaryland.edu; 6Creatv MicroTech, Inc., Rockville, MD 20850, USA; cmtang@creatvmicrotech.com; 7Department of Cell and Molecular Biology, School of Biological Graduate Studies, Perelman School of Medicine, University of Pennsylvania, Philadelphia, PA 19104, USA; ashvathi.raghavakaimal@pennmedicine.upenn.edu

**Keywords:** breast cancer, cancer-associated macrophage-like cells, beta-2 adrenergic receptor, circulating tumor cells, epithelial-to-mesenchymal transition cells

## Abstract

The usage of beta blockers in breast cancer (BC) patients is implicated in the reduction in distant metastases, cancer recurrence, and cancer mortality. Studies suggest that the adrenergic pathway is directly involved in sympathetic-driven hematopoietic activation of pro-tumor microenvironmental proliferation and tumor cell trafficking into the circulation. Cancer-associated macrophage-like cells (CAMLs) are pro-tumor polynucleated monocytic cells of hematopoietic origin emanating from tumors which may aid in circulating tumor cell (CTC) dissemination into the circulation. We examined the linkage between Beta-2 adrenergic receptor (B2AR) signaling in CAMLs and CTCs by establishing expression profiles in a model BC cell line (MDA-MB-231). We compared the model to CAMLs and CTCs found in patents. Although internalization events were observed in patients, differences were found in the expression of B2AR between the tumor cell lines and the CAMLs found in patients. High B2AR expression on patients’ CAMLs was correlated with significantly more CAMLs in the circulation (*p* = 0.0093), but CTCs had no numerical relationship (*p* = 0.1565). High B2AR CAML expression was also significantly associated with a larger size of CAMLs (*p* = 0.0073), as well as being significantly associated with shorter progression-free survival (*p* = 0.0097) and overall survival (*p* = 0.0265). These data suggest that B2AR expression on CAMLs is closely related to the activation, intravasation, and growth of CAMLs in the circulation.

## 1. Introduction

The usage of beta blockers in breast cancer (BC) patients has been shown to potentially reduce metastasis and recurrence, increase disease-free intervals, and prolong overall survival in patients with metastatic breast cancer [1]. Recently, several studies have observed that BC patients prescribed beta blockers have significant improvements in outcome. A large population-based study demonstrated that patients prescribed propranolol, a beta-1/beta-2 antagonist, were less likely to have larger tumors and lymph node involvement [2]. Furthermore, these patients also had a significantly lower cumulative probability of breast cancer-specific mortality [2]. A similar finding was also reported in a single-center retrospective study, which showed that patients treated with beta blockers had a reduction in metastases and a reduction in breast cancer-specific mortality [3]. Propranolol is a beta blocker drug which significantly decreases tumor size and nodal spread and improves overall survival (OS) in BC [2]. Lastly, inhibition of beta-adrenergic receptors has been shown to reduce the incidence of brain metastases in triple-negative BC [4]. The role of adrenergic signaling appears to aid in the progression and spread of BC, while targeted inhibition of this receptor appears to improve patient outcome. As the potential usage of beta blockers may benefit certain BC patients, better methods to distinguish these patients become crucial. One potential method may involve measuring the cellular expression of the adrenergic receptors on the cells disseminating from the tumor site.

Beta-2 adrenergic receptor (B2AR) is a member of the seven-transmembrane receptor G protein-coupled receptors (GPCRs), which is activated by stress via catecholamines, such as epinephrine and norepinephrine [5]. B2AR leads to downstream upregulation of cyclic AMP (cAMP) and further activation of protein kinase A (PKA), which can lead to the activation of cytokine production, growth factors, cell morphology, and cell death [6]. A sign of activation of B2AR is the internalization of the receptor. This process allows for either the clearance of ligands for resensitization of the receptor for further signaling or the potential lysosomal degradation of the receptor in a downregulation process [7]. Imaging of this activation and internalization of B2AR has been shown in various cell lines, including MB-231, by induction using the B2AR agonist isoproterenol [8,9,10]. Notably, the activation of B2AR has also been established in stromal immune cells, such as macrophages and other myeloid-derived niche cells, which appear to aid in the secretion of pro-inflammatory and pro-tumorigenic signals within tumors [11,12,13]. Further, B2AR activation causes migration and differentiation of macrophages at tumor sites. These tumor-associated macrophages can block cytotoxic T cells and inhibit therapeutic treatments from entering the tumor site [14,15]. B2AR is highly involved in the tracking of monocytic tumor-associated immune cells, whose quantification may provide biological information on the metastatic potential of specific pro-tumorigenic microenvironmental cells [13]. 

Cancer-associated macrophage-like cells (CAMLs) are a specific subtype of pro-tumorigenic giant polyploid myeloid cells that circulate in cancer patient blood, which derive from the tumor stromal microenvironment [16,17,18,19,20,21]. CAMLs express CD45 and CD14 but are distinguished by their polyploid nucleus and their large cell size [18]. Recently, CAMLs have been shown to present with hematopoietic stem-like and proangiogenic phenotypes whose high cell numbers and phagocytic engorgement (i.e., size enlargement) have been associated with poor patient outcomes [20,21,22,23]. While CAMLs have been theorized to be hematopoietic myeloid immune cells involved in cancer spread, their activation pathway has not been elucidated. In this prospective study, we evaluated the number of CAMLs, B2AR expression, and clinical outcomes in 31 late-stage BC patients’ blood samples for CAML enumeration and analysis with the B2AR expression profile to determine the relationship with CAML presence and clinical outcomes.

## 2. Materials and Methods 

### 2.1. Cell Culture and Bioassay

The MB-MDA 231 cell line was purchased from the ATCC and cultured to full confluence in accordance with ATCC guidelines in Leibovitz (L-15) Media with the addition of 10% FBS in O_2_ incubation. Cells were trypsinized and plated on two 8-well plates at 1 × 10^5^ cells/well (Nunc). After an overnight attachment period, the media were aspirated, and cells were treated with (A) media alone, (B) media + 2 μM of isoproterenol for 15 min, (C) media + 2 μM of isoproterenol for 30 min, or (D) media + 2 μM of isoproterenol for 60 min. The second plate was then treated as follows: (A) media alone as untreated control, (B) media + 1 μM isoproterenol for 1 h, (C) media + 2 μM isoproterenol for 1 h, (D) media + 5 μM isoproterenol for 1 h, (E) media + 50 μM isoproterenol for 1 h. (All experiments were run in replicates of four following the above protocol.) At each experimental endpoint above, wells were aspirated, and 250 μL of 1% paraformaldehyde in PBS was added to each well for 15 min. After 15 min of exposure, the wells were washed with 1X PBS and replaced with 250 μL of 1% Tween-20 for an additional 15 min, followed by 250 μL of blocking in 1XPBS + 2% BSA. Wells were washed with 1XPBS and stained according to the manufacturer’s protocol for immunofluorescence with 1:200 dilution of B2AR antibody (Cell Signaling clone (D6H2)) for 1 h. Wells were then washed with 5mL 1XPBS + 0.1% Tween-20 and mounted using Flouromount-G with DAPI (Southern Biotech, Birmingham, AL, USA). Cells (n = 10 cells) were measured for each well for all four replicates of the experiment. Average fluorescence intensity and standard deviation measurements were recorded for both time-gated and dose-escalation experiments. 

### 2.2. Cohort Recruitment

This single-blind prospective pilot study was initiated to analyze the expression of B2AR on CAMLs found in the peripheral blood of late-stage BC patients. Blood samples were procured from 31 late-stage (stage III or IV) BC patients through collaboration efforts with Northwestern University, the Mayo Clinic Cancer Center, University of Maryland, and Thomas Jefferson University, in accordance with the local institutional review board (IRB) approval and with the patients’ informed written consent. BC patients were recruited from November 2015 through October 2020. Inclusion criteria for this study included patients with late-stage BC above the age of 18 (minors were not included in this study). For enumeration analysis, 7.5 mL blood samples were obtained from BC patients, collected using CellSave Preservation Tubes (Menarini-Silicon Biosystems), anonymized, shipped, and analyzed within 96 h for CAML enumeration and B2AR expression at the Creatv MicroTech facility. Patients’ medical status was monitored for 24 months after treatment induction with clinical variables, including age, gender, resectability, pathological staging when available, and all neoadjuvant and adjuvant therapeutics being evaluated (Table 1).

### 2.3. Analysis of Filters

Patient samples were filtered using a CellSieve^TM^ microfilter (Creatv Microtech, Rockville, MD, USA), as previously described. Samples were prefixed for 15 min prior to filtration, followed by CellSieve™ microfiltration using a vacuum at a pre-calibrated pressure. Blood samples were further treated with post-fixation and permeabilization buffer and stained using 1:200 B2AR monoclonal antibody stain (Cell Signaling). Briefly, the CTCs, EMTs, and CAMLs were evaluated based on morphological features and expression of other biomarkers such as cytokeratin and CD45. CAML expression profiles for markers such as cytokeratin, CD45, CD14, PDL-1, and B2AR were included (Appendix A). For standardization, the background intensity was subtracted from the cellular intensity. Filter analysis was conducted using an Olympus BX51WI fluorescent microscope with imaging using a Carl Zeiss AxioCam monochrome camera for CAML enumeration (Figure 1). The Zen2011 Blue program was used for the processing of the images. Cell size and intensity measurement tools were analyzed using pre-calibrated Zen2011 Blue software (Zen 2.3 Blue edition, Carl Zeiss Microscopy GmbH, White Plains, NY, USA). 

### 2.4. Statistical Analysis

Univariate analysis of B2AR expression on CAMLs was conducted using Cox proportional hazard regression at a statistical threshold of *p* ≤ 0.05, using MATLAB R2020. Comparison between CAML number and size in high- and low-B2AR-expressing groups was performed using Wilcoxon rank-sum analysis (Figure 1). Progression-free survival (PFS) and overall survival (OS) Kaplan–Meier estimations were performed using the time to progression and mortality, defined as the time interval between the first induction of treatment and the occurrence of an event, by standard RECIST criteria using PET/CT scans or death, within a 2-year endpoint. 

## 3. Results

MDA-MB-231 cell lines were used to optimize and model B2AR expression and observe its potential activation and internalization in this polynucleated cell line model. We treated the cell lines with varying concentrations of isoproterenol for 60 min to induce activation of the adrenergic pathway (Figure 2 and Appendix A). Average B2AR expressions were measured without the addition of isoproterenol or with an introduction using concentrations of 1 μM, 2 μM, 5 μM, and 50 μM over 60 min. It was observed that, prior to treatment with isoproterenol, B2AR expression in MDA-MB-231 mainly resided on the cell surface when grown in standard L-15 media with an average fluorescence intensity of 314 (Figure 2a,b,g). Once exposed to 1 μM isoproterenol for 1 h, the average intensity non-significantly rose to 465 with the observed presence of dotting, theorized to be an example of internalization of the B2AR marker (*t*-test, *p* = 0.175) (Figure 2a,c,h) [12]. This trend continued with increasing concentration to 2 μM, showing both internalization events and an increased average fluorescence intensity of 543 (*t*-test, *p* = 0.496) (Figure 2a,d,i). The average intensity reached a peak with the introduction of 5 μM, with an average intensity of 858 and significantly increasing observable internalization events (*t*-test, *p* = 0.027) (Figure 2a,e,j). Somewhat paradoxically, when isoproterenol was introduced at a concentration of 50 μM, there was a non-significant decline in the average intensity to 591 (*t*-test, *p* = 0.079) (Figure 2a,f,k). This may be due to an oversaturation of the B2AR receptor, as most of the B2AR appeared to have been internalized into the cell, as found in the previous literature [12,24]. Further, the dose-escalation bioassay run in conjunction evaluated the effects of isoproterenol at various exposure times of the optimized 2 μM concentration of 0 min, 15 min, 30 min, and 60 min. In this gated experiment, the average intensity quickly peaked within 15 min of exposure, with an average intensity of 640 (Appendix A). The intensity appeared to plateau between the 15 min and 30 min exposure times, as the average intensity was once again ~640 (Appendix A). After 60 min of exposure, the average intensity dropped ~15% to 548, suggesting the bioassay reached saturation (Appendix A). These results suggest that prolonged B2AR signaling can result in upregulation of B2AR on a cell’s surface and may result in increased internalization of the receptor. 

We acquired a total of 42 blood samples from 31 individual patients, with 8 patients having serial follow-up blood samples for analysis (Table 1). Among the 31 patients, 30 patients had stage IV disease, and 1 patient had stage IIIB. The distribution of histology was 58.1% (n = 18/31) invasive ductal carcinoma, 6.5% (n = 2/31) invasive lobular carcinoma, 6.5% (n = 2/31) inflammatory breast cancer, and 29.0% (n = 9/31) having an unknown histology. The mean age was 57 years old (range 35–84 years). At the end of the 2-year observation period, (n = 24/31) patients progressed, with (n = 5/31) patients censored and (n = 2/31) patients lacking survival information. Further, (n = 11/31) patients experienced death within 2 years, with an additional (n = 18/31) patients being censored. 

In contrast to the results of the cell line experiments, a threshold of 65 was optimal in stratifying the patient population into a high and low grouping. A total of 14 out of 42 samples had low B2AR expression, and 28 out of 42 samples had high B2AR expression. The median number of CAMLs was seven CAMLs/sample in all blood samples, and the median CAML size was 120 microns. Further, it was determined that samples with high B2AR expression on the CAMLs had a significantly higher number of CAMLs than samples with low B2AR expression on the CAMLs (19 vs. 4 CAMLs/samples, *p* = 0.0093; Figure 3a). Moreover, patients with high B2AR expression on the CAMLs had significantly larger CAMLs than patients with low B2AR expression on the CAMLs (151 μm vs. 62 μm, *p* = 0.0073; Figure 3b). However, B2AR measurements on the CTCs were not possible as only 13 out of 42 samples had CTCs in circulation, and only 2 samples had CTCs available for B2AR analysis. Further, a higher expression of B2AR on the CAMLs did not correlate with increased numbers of CTCs (*p* = 0.1565) or EMTs (*p* = 0.5527) found in the circulation (Appendix A). 

Once relationships between B2AR expression on the CAMLs and their number and size were identified, we monitored the patients’ PFS and OS over 2 years. Overall, patients with low B2AR had an increased median PFS (mPFS) of 10.9 months and a median OS (mOS) of 13.4 months. In contrast, in patients with high B2AR expression, the mPFS was 3.0 months and the mOS was 5.5 months. In addition, logrank analysis of 24-month PFS found that high-B2AR-expressing patients progressed at significantly faster rates (HR = 3.5, 95% CI = 1.5–8.7, *p* = 0.0097) (Figure 4a) and had significantly faster rates of mortality (HR = 5.0, 95% CI = 1.4–17.2, *p* = 0.0265) (Figure 4b). 

## 4. Discussion

In this study, we optimized and quantified the expression of B2AR on the model breast cancer cell line MDA-MB-231 for activation, expression, and internalization of a well-established sympathetic nervous system hormone/neurotransmitter-based receptor. To accomplish this, two separate bioassays were run in conjunction to test (1) the dose-escalation effects of isoproterenol and (2) the effects of exposure time to the chemical on the expression of the adrenergic receptor. Our cell line models demonstrated two types of B2AR expression profiles, upregulation in expression in response to B2AR stimulation, and internalization seen in the form of dotting. These various expression profiles acted as an observable model for analyzing and quantifying increased B2AR stimulation in patients’ circulating cells. In the 42 samples from the 31 BC patients, different expression profiles and similar dotting were believed to be internalization found in patients’ CAML samples (Figure 1). This appears to suggest adrenergic pathway signaling within CAMLs and may support the hypothesis that CAMLs originate from stem niches in the body. 

We based our initial cell line experiments on the literature involving B2AR activation and the use of agonists such as isoproterenol [25,26]. In the presence of isoproterenol, many MDA-MB-231 cells formed pools of B2AR (Figure 2c–f,h–k). These pools of B2AR formed quickly after exposure to isoproterenol, with collections appearing 15 min post-exposure to the agonist (Appendix A), which corresponds to the prior literature [25]. An interesting observation was made in the time-sensitive bioassay (Appendix A). Between 15 and 30 min, the intensity of the signal plateaued (Appendix A), indicating that the upregulation of B2AR occurred early post-exposure to the agonist and remained at similar intensities after this initial rise in expression. In addition, after 60 min of exposure to isoproterenol, B2AR appeared to decline in expression intensity. This may be due to the competitive inhibition resulting from receptor internalization. As receptors become ligand-bound, the cell internalizes the receptor to resensitize it for future signaling. Theoretically, as more B2AR pools together and becomes internalized—a process that leads to resensitizing and recycling the receptor back to the cell surface—the cell surface intensity of B2AR would logically decline during this process. Also of interest was the effect of dose escalation on B2AR in response to increasing isoproterenol concentrations. As greater concentrations of isoproterenol were used, the average intensity of B2AR rose greatly in expression (Figure 2a). This rise in expression persisted until a 50 μM concentration of the beta-adrenergic agonist was added. This may be due to the oversaturation of the agonist, causing much greater internalization into the MDA-MB-231 cells, resulting in a lower cell surface B2AR intensity. Although not quantifiable in this experiment design, greater pooling and internalization events were observable in the 50 μM cell lines (Figure 2f,k). Similar to the exposure time assay (Appendix A), we hypothesized that as more B2AR is internalized, less surface signals and lower expressions are measured. 

Within patient samples, increased B2AR expression on CAMLs appeared to show a significant positive correlation with increased numbers of CAMLs found in the circulation (Figure 3a), as well as a significant positive correlation with larger CAMLs found in the circulation (Figure 3b). Further, the higher expression of B2AR appeared to be predictive of both PFS (*p* = 0.0097) and OS (*p* = 0.0265) (Figure 4). These results suggest a relationship of adrenergic signaling in BC to progression and aggressiveness of disease as they relate to these circulating cell populations. However, these results are limited by the size of the patient cohort (n = 31 BC patients), requiring larger, more comprehensive studies to better understand the relationship of B2AR to CAMLs, and possibly their potential as it relates to B2AR-targeted therapies. 

## 5. Conclusions

Increased tumor-associated macrophage (TAM) invasion in breast cancer can have immunosuppressive and pro-tumorigenic effects [13]. As the proportion of stromal tissue occupied by TAMs increases, less cytotoxic T cells and treatments are able to reach the innermost tumor cells; therefore, it has been suggested that decreasing monocyte/macrophage migration to neoplastic sites is vital for advancing treatment. As increased B2AR signaling has been suggested to have activation and migratory effects in stromal macrophages within tumor microenvironments, it becomes important to better understand the underlying biologic effect of BC [6,11,15]. CAMLs have been shown to be circulating cells of myeloid stromal origin as they stain positive for markers CD14 and CD45 and emanate from primary tumors [17,27]. CAMLs are a defined subgroup of TAMs and may function similarly by preventing the entry of other immune cells and cytotoxic therapies. Therefore, it can be hypothesized that patients who have greater expression of B2AR on CAMLs may also have greater increased activation and intravasation, though this was not evaluated in this manuscript. The previous literature has described that increased adrenergic signaling in cancer can increase cancer growth [6,10,28]. Further, increases in tumor-associated macrophage activation have been suggested to increase the migration of these myeloid cells [14,29]. As such, it stands to reason that the increased B2AR signaling and activation in CAMLs may be related to the increased number in circulation. Although this study did not measure the catecholamine presence in the circulation, the presence of high expression of B2AR on CAMLs and the association with poorer clinical outcomes may imply a possible relationship between CAMLs, beta-adrenergic signaling, and patient progression. As the usage of beta-adrenergic antagonists decreases macrophage migration in tumor sites, it might be possible that B2AR expression on CAML migration may also be affected by these antagonists [15]. If these patterns hold, this could explain a potential biological mechanism of B2AR and migration of myeloid cells. If CAMLs are being activated by epinephrine/norepinephrine, causing increased migration to and from tumor sites, this might offer another potential site for therapeutic intervention as an immune-modulating response, though additional studies are required to evaluate this hypothesis. 

In this initial pilot study, high B2AR expression on CAMLs was significantly associated with shorter PFS and OS. B2AR has long been a theoretical target in the treatment of BC [6,30], and recent studies have shown the therapeutic effect of beta-adrenergic antagonists in decreased signaling, inhibition of pro-metastatic cellular and molecular pathways, and lowering disease recurrence [31]. Further, while there is a clear need to expand this study and validate these preliminary findings, it would be of interest to measure B2AR in CAMLs of patients undergoing drug therapies targeting adrenergic receptors, such as B2AR, which are currently being studied in the clinical trials NCT00502684 and NCT02013492. This may require monitoring B2AR expression in CAMLs throughout the course of treatment to determine its relationship with the drug effect. In this study, we did not evaluate the change in B2AR over time, though it would be of great interest to evaluate B2AR expression on CAMLs in its ability to predict response to anti-adrenergic therapies in late-stage BC patients over time. In summary, our pilot study demonstrated potential important roles of B2AR in CAMLs related to metastasis and disease progression. Future studies are needed to confirm this finding. 

## Figures and Tables

**Figure 1 ijms-23-07299-f001:**
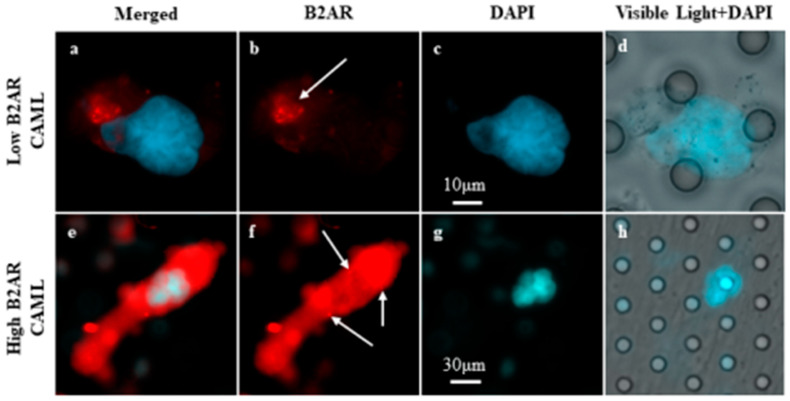
Images of low- and high-B2AR-expressing CAMLs. CAMLs are enlarged cells with a polyploid nucleus (blue) and can be B2AR-positive (red). (**a**–**d**) CAMLs showing low expression of B2AR (box = 60 µm, scale bar = 10 µm), with the white arrow pointing to potential internalization. (**e**–**h**) CAMLs showing high expression of B2AR (box = 180 µm, scale bar = 30 µm), with the white arrow pointing to potential internalization.

**Figure 2 ijms-23-07299-f002:**
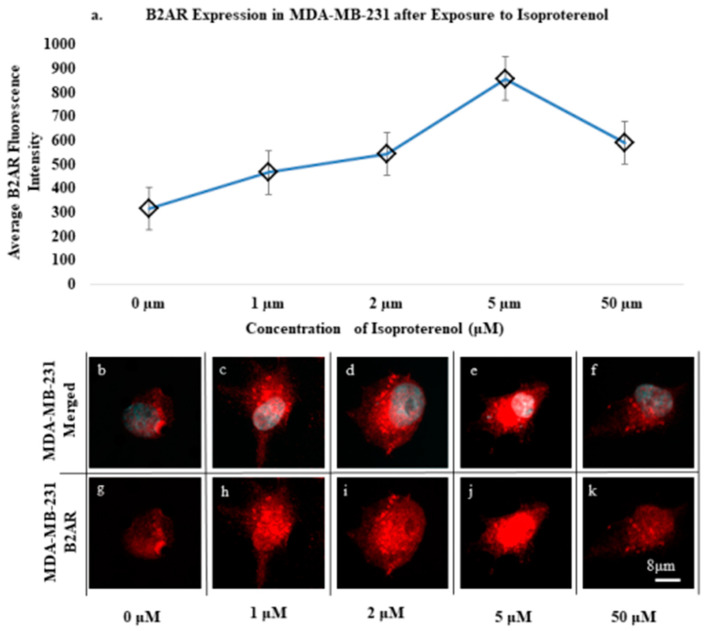
Average B2AR fluorescence intensity and endosome formation from increasing concentrations of isoproterenol: (**a**) Intensity of B2AR in the MDA-MB-231 cell line compared to increases in the concentration of isoproterenol. Bars = S.E. (**b**–**f**) Merged images of the MDA-MB-231 cells exposed to isoproterenol at 0, 1, 2, 5, and 50 μM for 60 min. (**g**–**k**) B2AR images of the MDA-MB-231 cells exposed to isoproterenol at 0, 5, 10, 20, and 50 μM for 60 min. More dotting is shown as endosome formation happens closer to the perinuclear space at increased concentrations of isoproterenol. (boxes = 45 µm, scale bar = 8 µm).

**Figure 3 ijms-23-07299-f003:**
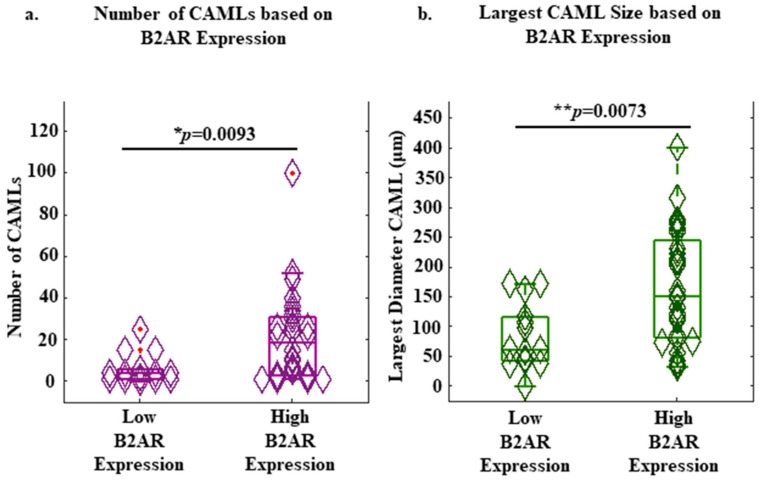
Relationship of B2AR expression of the CAMLs in relation to CAML number and the largest CAML in circulation. (**a**) Whisker plots of patients’ CAML numbers based on high/low expression of B2AR. Wilcoxon rank-sum (* *p* = 0.0093). Red dots = data outliers. (**b**) Whisker plots of the largest-diameter CAML from each patient’s sample based on high/low expression of B2AR. Wilcoxon rank-sum (** *p* = 0.0073).

**Figure 4 ijms-23-07299-f004:**
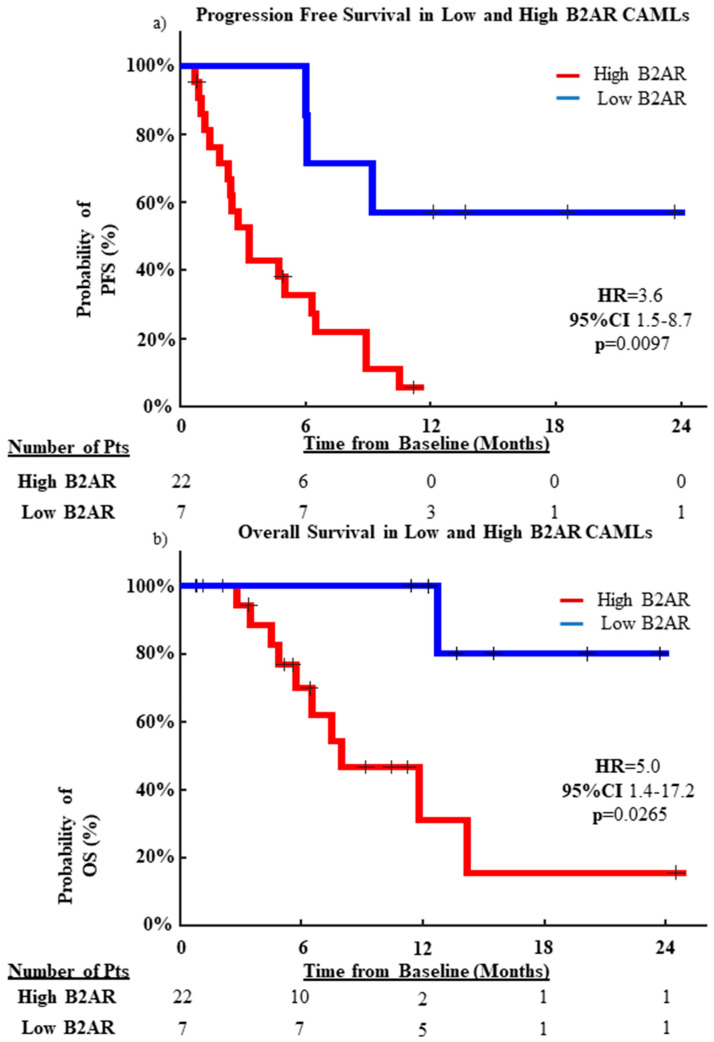
Kaplan–Meier graphs of PFS and OS for B2AR expression between high- and low-B2AR-expressing patients for total patient population. (**a**) PFS of patients with high-B2AR CAMLs (mPFS = 3.0 months) vs. low-B2AR CAMLs (mPFS = 10.9 months). (**b**) OS of patients with high-B2AR CAMLs (mOS = 5.5 months) vs. low-B2AR CAMLs (mOS = 13.4 months).

**Table 1 ijms-23-07299-t001:** Demographic table of the patients used in this study.

**Total Number**	(n = 31)
**Age Median (Range)**	58.5 (34–84)
**Race**	
**Caucasian**	18 (58.1%)
**African American**	3 (9.7%)
**Asian American**	1 (3.2%)
**Hispanic**	1 (3.2%)
**Other/Unknown**	7 (22.6%)
**Histology**	
**Invasive Ductal Carcinoma**	18 (58.1%)
**Invasive Lobular Carcinoma**	2 (6.5%)
**Inflammatory Breast Cancer**	2 (6.5%)
**Unknown Histology**	9 (29.0%)
**Grade**	
**II**	5 (16.1%)
**III**	12 (38.7%)
**Unknown Grade**	14 (45.2%)
**Pathological Stage**	
**III**	1 (3.3%)
**IV**	30 (96.7%)
**ER+**	11 (36%)
**PR+**	6 (19%)
**HER2+**	3 (10%)
**TNBC**	12 (39%)

## Data Availability

The data for this study have been made available at: https://doi.org/10.6084/m9.figshare.20188145.v1 (accessed on 29 June 2022).

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
