# Peer review of "Beta 2-Adrenergic Receptor in Circulating Cancer-Associated Cells Predicts for Increases in Stromal Macrophages in Circulation and Patient Survival in Metastatic Breast Cancer"

_ijms, 2022, doi:10.3390/ijms23137299_

Round 1

Reviewer 1 Report

The authors studied the relationship between B2A signaling in CAMLs and CTSs. The paper is scientifically sound and has studied in human objects over 2 years. Even though the topic itself is not novel but due to the high importance of the breast cancer, I recommend to publish this paper as it is.

Author Response

We appreciate the reviewer’s constructive comments. We have included our point-by-point response to each critique and have made changes listed in red in the revised manuscript attached. Below, please find each critique brought up by the reviewers is in italic and our responses are in bold:

Reviewer 1: “The authors studied the relationship between B2A signaling in CAMLs and CTSs. The paper is scientifically sound and has studied in human objects over 2 years. Even though the topic itself is not novel but due to the high importance of the breast cancer, I recommend to publish this paper as it is.”

  1. We thank the Reviewer for their kind words and support for the publication of the manuscript.

The above is taken from the cover letter, addressing all reviewers point-by-point, attached below in a word document.

Reviewer 2 Report

- Title: the term "recruitment" is not appropriate since the study does evaluate the expression level of B2AR only in circulating CAMLs

-CAMLs sorting/filtering should be better detailed. Specific CAML markers should be also evaluated and reported in the manuscript.

-Figure 3,b. The title of the y axis should be better specified. Largest CAML is not a physical quantity.  

Lines 227-228: the sentence is not clear. Was the classification of CAMLs into low or high B2AR expression based on the MB-231 model or on what was reported in Figure 1?

-The title of section 4 should be changed in discussion. Moreover, this part in highly speculative and should be rewritten.

Author Response

            We appreciate the reviewer’s constructive comments. We have included our point-by-point response to each critique and have made changes listed in red in the revised manuscript attached. Below, please find each critique brought up by the reviewers is in italic and our responses are in bold:

Reviewer 2: “Title: the term "recruitment" is not appropriate since the study does evaluate the expression level of B2AR only in circulating CAMLs

  1. We fully agree with the opinion of the reviewer. We have changed the title to “Beta 2-adrenergic receptor in circulating cancer associated cells predicts for increases of stromal macrophages in circulation and patient survival in metastatic breast cancer”.

Reviewer 2: “CAMLs sorting/filtering should be better detailed. Specific CAML markers should be also evaluated and reported in the manuscript.”

  1. We have included a more detailed explanation for the filtration technique in the methods section. This new section better details the process of CAML capture and staining.
  2. As requested, we have also included four various subtyping cell markers that were evaluated on the CAMLs along with the B2A. This data is now included as “Supplemental Figure 1 (Sup Fig 1)”.

Reviewer 2: “Figure 3,b. The title of the y axis should be better specified. Largest CAML is not a physical quantity.”

  1. We have changed the axis title to “Largest Diameter CAML (um)”.

Reviewer 2: “Lines 227-228: the sentence is not clear. Was the classification of CAMLs into low or high B2AR expression based on the MB-231 model or on what was reported in Figure 1?

  1. The patient CAMLs stratification into high vs low was not based on the cell line results. To clarify, we have edited this section to state that “In contrast to the results of the cell line experiments, a threshold of 65 was optimal in stratifying the patient population into a high and low grouping..”.

Reviewer 2: “The title of section 4 should be changed in discussion. Moreover, this part in highly speculative and should be rewritten

  1. We apologize for the confusion; we had merged these sections in error. We have now separated the conclusion and discussion into two sections. Further, we have re-written the speculative portions of the discussion to emphasize the need for additional experiments to prove any specific hypotheses.

The above is taken from the cover letter, addressing all reviewers point-by-point, attached below in a word document.

Reviewer 3 Report

Dear authors!

The image of the cells in transmitted light should be added to Figure 1 as well as the scale.

What do the authors mean by "overal cell intensity" in Figure 2? If fluorescence intensity is being assessed, then this should be indicated. How was the data in Figure 2A obtained and what do they represent? If it is an average, how many cases was it obtained from? This must be indicated in the signature. There is no indication that an assessment of statistical significance has been made between the data in Figure 2A. This information must be added.

The authors ignored the Discussion section, but discussed their results in the Conclusions. This misleads the reader. It is necessary to summarize the conclusions in the Conclusions and discuss the results in the Discussion, which meets the requirements of the journal.

In general, the manuscript contains little data obtained at the molecular level, and therefore poorly corresponds to the subject of the journal, in my opinion. Of course, the data obtained by the authors are interesting from both fundamental and practical points of view.

Author Response

            We appreciate the reviewer’s constructive comments. We have included our point-by-point response to each critique and have made changes listed in red in the revised manuscript attached. Below, please find each critique brought up by the reviewers is in italic and our responses are in bold

Reviewer 3 “The authors ignored the Discussion section, but discussed their results in the Conclusions. This misleads the reader. It is necessary to summarize the conclusions in the Conclusions and discuss the results in the Discussion, which meets the requirements of the journal.”

  1. We apologize for the confusion; we had merged these sections in error. We have now separated the conclusion and discussion into two sections. Further, we have re-written the speculative portions of the discussion to emphasize the need for additional experiments to prove any specific hypotheses.

Reviewer 3: “The image of the cells in transmitted light should be added to Figure 1 as well as the scale.

  1. We have added an image of visible light overlapped with DAPI as “Figure 1d and Figure 1h” as requested. We have also added a scale to both cells in Figure 1.

Reviewer 3: “What do the authors mean by "overal cell intensity" in Figure 2? If fluorescence intensity is being assessed, then this should be indicated. How was the data in Figure 2A obtained and what do they represent? If it is an average, how many cases was it obtained from? This must be indicated in the signature. There is no indication that an assessment of statistical significance has been made between the data in Figure 2A. This information must be added.”

  1. The figure was the average CAML fluorescence intensity for each of the different concentrations of isoproterenol added. We have changed the axis to “Average B2AR Fluorescence Intensity” to clarify.
  2. We have now included the description of the experimental design for the cell line experiments in the Method section. The results were analyzed using 2 separate plates of MB-231 cells run with 1) varying concentrations of isoproterenol and 2) varying exposure times to isoproterenol. All plates were run in replicates of four. The average intensities were measured from approximately 10 cells per each well. These details are now added into the methods section.
  3. We have now included “T-test” and the “p values” for the changes in B2AR expression in the dose escalation experiments. We found that the increase to 5 μM was significantly different, compared to the other concentration changes of isoproterenol which were not significant.

Reviewer 3: “In general, the manuscript contains little data obtained at the molecular level, and therefore poorly corresponds to the subject of the journal, in my opinion. Of course, the data obtained by the authors are interesting from both fundamental and practical points of view.”

  1. This was a special edition in “Liquid Biopsy” and we felt the contents fit the overall requirements. But we thank the reviewer for agreeing that the manuscript was both interesting and scientifically practical.

The above is taken from the cover letter, addressing all reviewers point-by-point, attached below in a word document.

Round 2

Reviewer 2 Report

Authors addressed all the points raised by the reviewer. The paper is now acceptable for publication.

Reviewer 3 Report

The manuscript can be published.